# Health Literacy and Self-Care in Patients with Chronic Illness: A Systematic Review and Meta-Analysis Protocol

**DOI:** 10.3390/healthcare12070762

**Published:** 2024-03-31

**Authors:** Camilla Elena Magi, Stefano Bambi, Laura Rasero, Yari Longobucco, Khadija El Aoufy, Carla Amato, Ercole Vellone, Guglielmo Bonaccorsi, Chiara Lorini, Paolo Iovino

**Affiliations:** 1Department of Health Sciences, University of Florence, 50134 Florence, Italy; stefano.bambi@unifi.it (S.B.); laura.rasero@unifi.it (L.R.); yari.longobucco@unifi.it (Y.L.); khadija.elaoufy@unifi.it (K.E.A.); carla.amato@unifi.it (C.A.); guglielmo.bonaccorsi@unifi.it (G.B.); chiara.lorini@unifi.it (C.L.); paolo.iovino@unifi.it (P.I.); 2Department of Biomedicine and Prevention, Faculty of Medicine, University of Rome “Tor Vergata”, 00133 Rome, Italy; ercole.vellone@uniroma2.it; 3Department of Nursing and Obstetrics, Wroclaw Medical University, 50-367 Wroclaw, Poland

**Keywords:** chronic illness, chronic disease, non-communicable disease, self-care, health literacy, digital health literacy, protocol, systematic review, meta-analysis

## Abstract

Self-care plays a critical role in symptom recognition, management, and risk factor modification for patients with chronic illnesses. Despite its significance, self-care levels in this population are generally poor. Health literacy (HL) is pivotal for promoting effective self-care, yet the association across specific chronic illnesses remains fragmented and conflicting. Therefore, a systematic review and meta-analysis will be conducted. Inclusion criteria encompass quantitative studies involving adult patients with at least one chronic illness reporting on the association between a measure of HL and one or more elements of self-care behaviors as outcomes. Databases to be searched include PubMed, CINAHL, APA PsycINFO, Embase, Web of Science, and Cochrane Central Register of Controlled Trials. The studies will undergo risk of bias and certainty of evidence assessment using ROBINS-E and GRADE. Extracted data will include authors, publication date, aim(s), study location, design, sample characteristics, chronic illness type, study length, HL, and self-care measures. Understanding the link between HL and self-care can aid healthcare providers in implementing strategies to enhance health-promoting behaviors, contributing valuable insights to the scientific community and fostering nuanced discussions. This protocol ensures methodological transparency, stimulates discourse, and paves the way for informed interventions to improve overall health outcomes.

## 1. Introduction

Chronic illnesses are pervasive and persistent on a global scale, thus posing significant challenges to public health and, significantly, contributing to over 70% of global mortality [1]. Chronic illnesses encompass a spectrum of conditions, including cardiovascular diseases, cancer, chronic respiratory diseases, and diabetes [1]. The intricate etiology involves multifaceted factors such as genetic predispositions, physiological intricacies, environmental influences, and behavioral determinants [1].

The protracted nature of these conditions gives rise to a myriad of adverse outcomes, ranging from prolonged hospitalizations and elevated mortality rates, to disability and decline in the quality of life, which contribute to escalating healthcare costs [2,3]. 

Self-care has emerged as a multifaceted component in the management of chronic illnesses because it is significantly associated with a spectrum of positive outcomes in these patients [4]. According to the Middle Range Theory of Self-Care of Chronic Illness, self-care is defined as a group of behaviors focusing on the promotion of good health and treatment adherence (self-care maintenance), attentiveness to body and symptom recognition (self-care monitoring), and response to signs and symptoms when they occur (self-care management) [5,6]. 

Consistent engagement in self-care practices has been associated with illness stability, improved well-being, and higher quality of life [4,5]. Moreover, constant self-care monitoring allows for timely recognition of changes in signs and symptoms, thereby facilitating proactive management of the disease [4,5], culminating in significant reduction in symptom burden and mortality [5,6]. Despite the importance of self-care to improvement of outcomes, individuals affected by chronic illnesses often face significant challenges in maintaining adequate self-care practices [7]. This problem has prompted researchers to focus on possible predictors of self-care behaviors, with the hope of targeting them; among these predictors, health literacy (HL) is gaining prominence in recent years [8].

HL, including digital health literacy (e-HL), is defined as a set of skills for accessing, comprehending, evaluating, and applying health information [9,10]. HL and e-HL have emerged as powerful predictors of self-care behaviors in the context of chronic illness [9,10]. These skills not only empower individuals to navigate the complexities of illness management but also to make informed decisions and adopt healthier behaviors [11]. For example, targeting HL can be promising for strengthening smoking cessation, improving physical exercise, and adhering to medication regimens [12,13]. In turn, this approach is useful to improve distal outcomes, including mortality rates, healthcare service utilization, quality of life, and well-being [11,12,13]. 

The literature generally suggests a positive association between HL and self-care; specifically, higher HL levels predict enhanced self-care behaviors [14,15,16]. However, the existence of absent or negative associations is also acknowledged, as evidenced in specific studies such as the one published by Wong et al. [17], who enrolled patients with chronic kidney disease (CKD) and found that HL was not related to differences in treatment adherence or physical exercise. One additional aspect is the lack of clarity of possible underlying mechanisms explaining the relationship between HL and self-care. For example, a study by Du et al. [18] conducted a multiple mediation and found that self-efficacy and self-care agency could mediate the relationship between HL and health-promoting lifestyles in older adults. Another study by Chen et al. [19] found a negative association between HL and self-care management in patients with heart failure. Other studies highlighted psychological factors such as empowerment [20], engagement [21], degree of comprehension of health information [22], and intrinsic motivation [23]. However, in general, the existing literature does not invest in sufficient analytical methods to elucidate the underlying mechanisms, i.e., by limiting computation of correlational and regression methods. Finally, there is a dearth of evidence regarding potential moderators of the relation between HL and self-care. The existing knowledge so far points to educational attainment, cultural background, and accessibility to resources as possible moderating factors [24].

Overall, interest in the field of HL and self-care is increasing, but a systematic synthesis of the evidence is still lacking. To date, to the best of our knowledge, the only available review exploring these constructs in chronic illnesses has been published recently [25]; however, this work is limited in that self-care has only been operationalized as a reflection of medication adherence, thus neglecting its multidimensional nature. Having a systematic synthesis of the literature available on this topic enables collation of fragmentary evidence on the effect of HL on self-care behaviors, and achieves a more thorough understanding of the underlying mechanisms between these two constructs in chronic illness, which can inform public health decision-making, guide targeted interventions, and ultimately improve clinical outcomes [26,27]. 

Through this systematic review and meta-analysis, we will specifically investigate whether and to what extent HL is associated with self-care behaviors in patients with chronic illnesses. We will also examine the existence of factors mediating or moderating this relationship. This work also aims to integrate the most recent advancements in interventions and theories, ensuring the derivation of relevant and generalizable outcomes grounded in the Middle-Range Theory of Self-Care of Chronic Illness [5,28] and Health Literacy Skills Framework [11].

### 1.1. Theoretical Frameworks

To establish the groundwork for our systematic review and meta-analysis, we relied on the theoretical frameworks proposed by Riegel [5] and Squiers [11].

#### 1.1.1. Middle-Range Theory of Self-Care of Chronic Illness

The Middle-Range Theory of Self-Care of Chronic Illness [5], proposed by Riegel in 2012, defines self-care as a dynamic process encompassing health-promoting practices and illness management. This theory describes the three key concepts of self-care maintenance, self-care monitoring, and self-care management. Self-care maintenance involves the behaviors aimed at enhancing well-being and maintaining physical and emotional stability, including vigilance and adaptation. Some examples of these behaviors encompass medication adherence, physical activity, and a healthy diet. Self-care monitoring consists of a process of systematic and attentive body surveillance to detect signs and symptoms of the disease, for example, monitoring the side effects of medications, or degree of tiredness during daily activities. Self-care management involves responding to changes in physical and emotional signs and symptoms, for example, taking a medicine to relieve a symptom, or calling the providers for guidance. The theory is grounded in three assumptions and seven testable propositions, providing a comprehensive framework for understanding self-care in patients with chronic illnesses. The outcomes of self-care include disease stability, well-being, perceived control, and broader impacts such as reduced hospitalizations and costs. Factors like experience, motivation, and support influence self-care, with practical implications involving tailored interventions based on specific patient challenges. 

The Middle-Range Theory of Self-Care of Chronic Illness offers a structured approach for clinical practice and research, aiding in developing self-care measures and intervention studies. Its integration into systematic reviews improves article selection and addresses research questions effectively. By distinguishing maintenance, monitoring, and management, the theory aids in formulating inclusion and exclusion criteria and ensures accurate result relevance assessment. Additionally, it highlights factors like experience and motivation, suggesting potential subgroups or contexts. This integration enhances article selection accuracy, refines the theoretical framework, and contributes to conceptual coherence and methodological rigor. Incorporating the HLS framework is essential for advancing scientific understanding.

#### 1.1.2. Health Literacy Skills

The Health Literacy Skills (HLSs) conceptual framework [11], developed by Squiers in 2012, is a significant contributor to understanding the interplay between HL competencies, behaviors, and outcomes. Rooted in different previous HL theories, this framework specifically describes the factors related to the development of HL skills and the mediators between these skills and health outcomes. Factors related to HL skills are demographic characteristics, individual resources, capabilities, and level of prior knowledge. Mediators can be social support, motivation, or healthcare. To convey the effect of HL skills on outcomes there must also be appropriate comprehension of stimuli, for example, the conversation content with a doctor, brochures, and prescription labels.

HLS provides an exploration of the dynamics within HL, addressing individual-level operations across four key components. Its unique contribution lies in unraveling the relationships among literacy, numeracy, communication, and information-seeking skills. Beyond theoretical value, HLS offers practical applications in public health interventions. Future research focusing on rigorous testing of causal pathways will refine HL research. Integrating HLS into a systematic review with meta-analysis establishes a robust foundation for evaluating research quality, identifying promising methodologies, and enhancing overall conceptual coherence and methodological rigor.

### 1.2. Aims

This systematic review and meta-analysis aims to collate evidence from studies that investigates the association between HL levels and self-care behaviors in individuals grappling with chronic illnesses. We specifically sought to address the following inquiries:
What is the nature and strength of the relationship between HL and self-care across specific chronic illnesses?What are the potential moderating and mediating factors of the relationship between HL and self-care?

## 2. Methods

### 2.1. Protocol Registration and Guidelines

This protocol adheres to the Preferred Reporting Items for Systematic Review and Meta-Analysis Protocols (PRISMA-P) checklist [29] and has been registered in PROSPERO (registration number: CRD42024488061). The systematic review and meta-analysis will be conducted following the Protocol Statement of the Preferred Reporting Items for Systematic Reviews and Meta-Analysis (PRISMA) guidelines [30].

### 2.2. Eligibility Criteria

The review questions were crafted employing the PEO framework, which delineates the following components: P for patient or population, E for exposure, and O for outcomes. In this context, “P” refers to individuals directly impacted by chronic health conditions, “E” encompasses the domain of HL, including both traditional and electronic aspects, while “O” pertains to the exploration of self-care behaviors. 

We will include adults (≥18 years old) diagnosed with at least one chronic illness among the nine conditions selected from the list published by the Office of the Assistant Secretary for Health (OASH) [31]. Specifically, we will include those with a high morbidity and prevalence globally, such as hypertension, coronary artery disease (CAD), arthritis, CKD, heart failure (HF), stroke, asthma, chronic obstructive pulmonary disease (COPD), and type 2 diabetes mellitus (T2DM). Other conditions that are asymptomatic (e.g., hyperlipidemia), psychiatric (e.g., schizophrenia), acute, and those that gain little benefit from self-care (e.g., dementia) will be excluded. 

We will operationalize self-care behaviors as the practices of self-care maintenance, monitoring, and management, consistent with the Middle-Range Theory of Self-Care of Chronic Illness [5]. We will consider both (i) generic self-care behaviors (e.g., ensuring adequate sleep, avoiding of tobacco smoke, and stress management), and (ii) disease-specific self-care behaviors (e.g., taking aspirin or other blood thinners, regularly monitoring blood sugar, weight, etc.) 

HL will be operationalized as the ability to obtain, understand, evaluate, and apply health information or digital health information in order to make informed decisions and maintain one’s well-being [11]. Studies will be selected if they include adults with at least one chronic illness, based on the OASH list [31], and examine the relationship between HL or e-HL and self-care behaviors. 

We will only include published observational studies (e.g., cohort studies, case-control studies, cross-sectional studies) and randomized controlled studies (RCTs) if the authors report the association between HL and self-care at baseline on the whole sample, and mixed-method studies will be included only if data from the quantitative component can be extracted separately. Due to resource constraints, only Italian- and English-language studies will be considered for inclusion. No temporal constraints will be applied because the key concepts of HL and self-care are not relatively new in health research. Specifically, self-care first appeared in 1956 [32], and HL was introduced in 1974 [33].

### 2.3. Search Strategy 

Six electronic databases will be searched including PubMed (NLM), CINAHL (EBSCO), APA PsycINFO (EBSCO), Embase (Ovid), Web of Science (Clarivate Analytics), and Cochrane Central Register of Controlled Trials (for quantitative studies only). These databases were chosen due to their comprehensive coverage of global research in the field of self-care and HL. The process of study identification will encompass a combination of electronic search strategies and manual methods. Electronic searches will involve the utilization of specific search terms within the selected databases. Concurrently, we will meticulously screen the citations of relevant studies to ensure a comprehensive approach. A detailed sample search strategy for PubMed is provided in the Appendix A.

### 2.4. Study Selection

Following the completion of the search, all identified references from each database will be imported into Rayyan [34], and any duplicate entries will be systematically removed. The refined list of references will then undergo a screening of titles and abstracts against the predetermined inclusion criteria for the review by a minimum of two independent reviewers. Subsequently, the full text of the selected citations will undergo a comprehensive assessment against the inclusion criteria by two or more independent reviewers. In cases where discrepancies arise among the reviewers, a resolution will be sought through thorough discussion or, if necessary, by involving an additional researcher. The outcomes of the search, the study inclusion process, and the rationale for excluding papers that do not meet the inclusion criteria will be meticulously documented in the final systematic review and meta-analysis. These details will be visually presented in the PRISMA flow diagram [30].

### 2.5. Data Extraction

Two independent reviewers will conduct the data extraction, capturing comprehensive details for each study. The extracted information will include author(s), publication year, aim(s), study location, sample characteristics (such as sample size and sociodemographic factors like age, sex, ethnicity, and educational level), type of chronic illness, study design, study duration, and HL measurement details, along with scores (mean, SD, or n, and %), self-care measurement details along with scores (mean, SD, or n, and %), main results, and assessment of risk of bias (Appendix A). To enhance clarity and facilitate synthesis, data from longitudinal and cross-sectional studies will be extracted and presented separately in two distinct descriptive tables. In cases where discrepancies emerge between reviewers during the data extraction process, resolution will be sought through discussion. If needed, a third reviewer will be involved to ensure consensus. Additionally, authors of the respective papers will be contacted to obtain any missing or supplementary data necessary for a comprehensive analysis. 

### 2.6. Data Synthesis

The synthesized data will be presented qualitatively in tables and narrative form. To assess inter-rater reliability (IRR), Cohen’s κ value will be calculated using SPSS v.25. The IRR is reported to reflect the degree of agreement among reviewers on the screening and inclusion process of articles in the systematic review. Studies providing effect sizes for the relationship between HL and self-care will be considered for quantitative meta-analysis. However, this approach is contingent on the availability of sufficient studies with acceptable methodological quality (i.e., high or moderate certainty of findings) reporting the strength of association, such as correlation coefficients, odds ratios, and standardized mean differences [35]. Additionally, if feasible, we intend to estimate the pooled prevalence of participants exhibiting low levels of HL and self-care. The R software’s “meta” package (version 4.3.3) will be employed for meta-analyses. Heterogeneity among studies will be assessed using Cochran’s Q test and the Higgins and Thompson’s I^2^ test, with an I^2^ value > 50% and a *p*-value < 0.05 indicating significant heterogeneity [36]. In the presence of substantial heterogeneity (I^2^ > 50%), the random-effects model will be utilized; otherwise, the fixed-effects models will be applied. Sensitivity analyses, such as leave-one-out, will be conducted for studies with I^2^ > 50% [37]. Subgroup analyses will be planned for outcomes with a significant number of included studies (>10) [38]. In addition, subgroup analyses stratified by age will be conducted, since age is a significant variable associated with HL and self-care [12]. Publication bias will be assessed through funnel plots and Egger’s regression symmetry test when appropriate [39]. Forest plots will be used to present the effect sizes. For studies for which quantitative meta-analysis is not feasible or in case the studies exhibit excessive heterogeneity, results will be presented in a narrative format [40]. 

### 2.7. Assessment of Risk of Bias

Risk of bias assessment will be diligently conducted in each study by two independent reviewers, utilizing the Risk Of Bias In Non-Randomized Studies of Exposures (ROBINS-E) tool tailored for observational studies [41]. Any discrepancies arising during the assessment will be resolved through discussion, reaching consensus between reviewers, or with the involvement of a third reviewer if necessary. The ROBINS-E tool comprises seven key domains, encompassing bias due to confounding, bias in the measurement of exposures, selection bias, bias among post-exposure interventions, bias due to missing data, bias in the measurement of outcomes, and bias in the selection of the reported result. Each domain will be assessed individually, and the cumulative evaluations will inform the overall judgment regarding the risk of bias in the study, categorized as low, some concerns, high, or very high.

### 2.8. Assessment of Certainty in the Findings

Certainty in the study findings will be systematically evaluated using the Grading of Recommendations, Assessment, Development, and Evaluation (GRADE) approach [42]. The GRADE system incorporates five critical domains: risk of bias, imprecision, inconsistency, indirectness, and publication bias. The quality of evidence will be categorized into four levels: high (4), moderate (3), low (2), and very low (1). High-level evidence necessitates a randomized, double-blinded study design without selection biases. In the context of observational studies, moderate evidence, indicating exceptionally robust evidence from unbiased studies, is considered the highest level of proof for an association. This comprehensive evaluation will enhance the reliability and applicability of the systematic review’s findings.

## 3. Discussion

This systematic review and meta-analysis critically evaluate the existing literature on HL and self-care in chronic illness management, aiming to clarify the relationship between HL and self-care practices across different health conditions and individual characteristics. The exploration of potential moderating or mediating factors aims to provide a more thorough understanding of the underlying mechanisms and strength of this association. Implications for future research and practice will highlight avenues for developing targeted interventions leveraging the positive impact of higher HL on self-care behaviors, with an emphasis on tailoring interventions across diverse chronic health conditions. This study will also potentially contribute to refining current theories and models related to HL and self-care in chronic illnesses.

Surrounded by the global health crisis attributed to chronic diseases [1,2], substantial investments and informed interventions are needed to mitigate the escalating global burden. The significance of this systematic review and meta-analysis is underscored by identified gaps in existing knowledge, recognizing the inherent complexity of diverse chronic health conditions and individual characteristics. Strategic allocation of resources for research and interventions is imperative, guided by evidence-based projections to aid policymakers in assessing intervention impacts and planning the requisite healthcare workforce. This investment aims to enhance global health outcomes and improve quality of life, considering not only mortality but also prolonged morbidity, which contributes significantly to escalating healthcare costs, compromising global quality of life. Hence, a comprehensive approach is imperative.

In our endeavor to comprehend and analyze the identified studies, we aim to elucidate the underlying mechanism linking HL and self-care behaviors. This entails a focused examination of potential moderating or mediating factors, which bear significant scientific implications. Moderation involves examining how the relationship between two variables changes depending on a third variable. Mediation explores the process by which one variable influences another one through an intervening variable, known as the mediator. In other words, the moderator is a variable that affects the relationship between two other variables, while the mediator explains the relationship between the predictor and outcome variables. The use of statistical moderation and mediation allows for a deeper understanding of the mechanisms through which HL impacts self-care. Concurrently, investigating moderation provides insights into contextual variations, determining whether the association between HL and self-care is contingent upon specific factors like the nature of the chronic condition or individual characteristics. This analytical approach not only contributes to theoretical refinement but also bears practical implications. Understanding mediating processes aids in designing targeted interventions by addressing specific pathways, while recognizing moderating factors facilitates the customization of interventions based on patient or contextual attributes. Consequently, this scientific endeavor has the potential to optimize interventions, enhance clinical decision-making, and advance the field of chronic illness management.

Methodologically, our approach is anchored in rigorous criteria for study selection, comprehensive data extraction, and thorough quality assessment. Recognizing and addressing potential challenges, including variations in study methodologies, is integral to ensuring the robustness and validity of our results [43]. This comprehensive exploration contributes to refining existing theories and models related to HL and self-care in chronic illnesses while adopting a patient-centered approach. The integration of patient perspectives aims to capture the holistic impact of HL on self-care behaviors, acknowledging the lived experiences of individuals managing chronic illnesses [44]. Our goal, in addressing implications for future research and practice, is to identify opportunities for developing targeted interventions that leverage the positive impact of higher HL on self-care behaviors. Emphasizing the diversity of chronic health conditions underscores the need for tailored interventions to ensure effectiveness across different patient populations [45]. 

Our systematic approach serves to refine existing theories and models while critically evaluating future study limitations. Integrating multidisciplinary insights enriches our study’s depth [46], aiming to address biases and gaps in current literature and provide a solid foundation for future research [47]. Clear objectives, robust methodologies, and ethical considerations are integral components. Anticipating impact on healthcare policy, we aim to inform targeted interventions to improve self-care outcomes for those with chronic illnesses. Disseminating our systematic review underscores transparency and scientific rigor [48,49], promoting reproducibility and scholarly discourse. This endeavor contributes to advancing effective strategies for managing chronic illnesses and reducing the global health burden, aligning with the imperative of addressing chronic diseases.

### Strengths and Limitations

This protocol has potential strengths and limitations that are worth mentioning. First, potential discrepancies between reviewers during screening and data extraction will be systematically addressed to mitigate the introduction of bias that could significantly impact overall reliability. The assessment of inter-rater reliability will intricately involve the utilization of the Kappa index, with concerted efforts to proactively address any variations in interpretation that may arise. Second, methodological heterogeneity among the studies included in our analysis poses a challenge due to inherent differences in study design, measurement tools, and participant characteristics. Acknowledging this heterogeneity will be essential for a clear interpretation of the findings.

Additionally, the reliance on published literature will be explicitly recognized as a potential source of publication bias, and careful attention will be devoted to understanding its potential influence on the study’s outcomes. Furthermore, the acknowledgment of generalizability limitations is paramount, considering the diverse nature of chronic illnesses and geographic variations present in the included studies. Variability in study quality, sample sizes, and control for confounding variables will be meticulously considered as factors that may significantly impact the strength of the evidence generated. The inclusive approach adopted in our study will be acknowledged as having the potential to hinder the drawing of specific conclusions for individual conditions, emphasizing the broader implications of our findings. Moreover, despite our best efforts to explore moderating factors, the presence of unaccounted confounding variables is expected to persist, influencing the underlined mechanism between HL and self-care behaviors.

## 4. Conclusions

In conclusion, this systematic review and meta-analysis aims to illuminate the relationship between HL and self-care behaviors among individuals with chronic illnesses. We emphasize HL as a critical factor influencing successful self-care practices. Our study has practical implications for healthcare professionals, policymakers, and educators, who can utilize these insights to develop targeted interventions aimed at improving HL levels and enhancing self-care behaviors in individuals managing chronic illnesses. This approach holds promise for reducing the burden of chronic diseases and improving overall health outcomes. Our commitment to transparency and scientific rigor is evident in the dissemination of our research protocol, fostering scholarly discourse within the scientific community. By contributing to the ongoing dialogue on HL and self-care, we strive to advance the development of effective strategies for managing chronic illnesses and reducing the global health burden associated with these conditions.

## Data Availability

Data are contained within the article.

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
