# Peer review of "Health Literacy and Self-Care in Patients with Chronic Illness: A Systematic Review and Meta-Analysis Protocol"

_healthcare, 2024, doi:10.3390/healthcare12070762_

Round 1
Reviewer 1 Report
Comments and Suggestions for Authors
The authors are to be congratulated with an interesting design for a protocol study of a systematic review and meta-analysis to investigate the relationship between Health literacy and self-care in patients with chronic illness. The theory behind the study rationale is mostly sound and can be developed further, while some parts may be shortened. The text is clear and easy to follow. The suggested methods to be applied are robust and adequate. Some terminology needs to be explained more.
There are several concerns with the study, see below.
Detailed comments
1 Introduction Line 64
"HL and e-HL have emerged as powerful predictors of self-care behaviors in the context of chronic illness [9,10]." Please explain how you can calin this since in reference [9] the term "self-care" or "self-care behaviors" is not mentioned. In reference [10] the term "self-care" is only mentioned on a few places. Is "health litteracy the same as self-care"? Is self-care the same as "self-management" and "self-efficacy"? Please explore more.
2 Line 110
reference Riegel [29] is a duplicate of Riegel [5]. Please correct.
3 Line 110-141-
A complete description of the theoretical framework The Middle-Range Theory of Self-Care of Chronic Illness is not necessary and takes to much space. This text needs to be shortened significantly, especially the latter part (line 130-141).
4 Line 164
Since you mention "strength of the relationship" in your aims, it is worthwhile to describe somewhere, e.g. under the Methods section, how you define "strength of the relationship".
5 Line 165
Since you mention "moderating and mediating factors" in your aims, it is worthwhile to describe already in the Introduction how you define "moderating factors" and "mediating factors". As "moderation" and "mediation" are used with very different definitions etc., there is a need that you describe these terms properly, and also how you want to investigate "moderating factors" and "mediating factors".
6 Line 291
Discussion: Please consider changing and shortening some parts of the Discussion. At some places the text is almost repeated, for example you mention more than one time that the study provides a solid foundation for future research, that you use robust methodology, that you target current knowledge gaps, etc. It is worthwhile that you mention these arguments, but pleas do not repeat your arguments.
7 Line 247
"To assess inter-rater reliability (IRR),..." Please clarify which data that are compared. Is it the result of the risk of bias assessment by both raters that is compard, or is it something else that is compared? Scale(s)?
8 Line 313-
"the identified studies will unravel the underlined mechanism between HL and self-care behaviors" I think this statement is a bit too strong, you cannot give this guarantee. Please amend your statement.
9 Line 381
Conclusion: Similar to the comment under the Discussion: please consider changing and shortening some parts of the Conclusion. In some parts the text is very similar and needs to be shortened.
Author Response
Thanks for the efforts in reviewing our manuscript. The remarks are interesting and thoughtful. We have addressed all of them accordingly. Please see the attachment.

Reviewer 2 Report
Comments and Suggestions for Authors
The systematic review protocol demonstrates a clear and detailed methodology, with well-justified approaches that are thoroughly explained, referenced, and effectively communicated. I have included some suggestions for further refinement that could strengthen the protocol. Overall, excellent work in developing a comprehensive and transparent framework for conducting the systematic review.
1. In line 35, under ‘Key words’, I suggest removing ‘nursing education’ from the list.
2. In line 103,the second aim states ‘We will also examine the existence of factors mediating or moderating this relationship’. Although mentioned in line 315 to 327 in the discussion section, I suggest to include definition or explanation of the terms ‘mediating factor’ and ‘moderating factors’ in the methodology section.
3. In line 104, the third aim ‘This work also aims to integrate the most recent advancements in inter- 104 ventions and theories, ensuring the derivation of relevant and generalizable outcomes 105 grounded in Middle-Range Theory of Self-Care of Chronic Illness [5,28] and Health Liter- 106 acy Skills Framework’ is not translated as a third aim under title ‘aim’ in line 160.In addition, this aim is not clearly manifested in the methodology section.
4. Under the title of ‘data extraction’:
- I suggest to mention a supplementary that includes your planned data charting instruments.
- In line 234, the extracted information list doesn’t include anything related to your second aim on mediating and moderating factor examination.I suggest including an extraction slot for mediating/moderating factor evidence.
Comments on the Quality of English Language
Minor editing of English language required
Author Response

(The authors gave the same response as above.)

Reviewer 3 Report
Comments and Suggestions for Authors
STRENGTHS
a) Good foundation for the paper in the Introduction section.
b) The way the theoretical framework is used by the authors is well explained; additionally, the use of the theoretical framework provides a solid basis for article selection and analysis.
WEAKNESSES
a) Sections 1.1.1 and 1.1.2, although being well explained, lack any citation to back up the statements made by the authors. Kindly address this.
b) PEO based research questions for systematic reviews are best used for qualitative studies. Can the authors please address this issue by providing a solid explanation for the use of PEO in their study where they used a quantitative approach? Thank you.
Author Response

(The authors gave the same response as above.)

Round 2
Reviewer 3 Report
Comments and Suggestions for Authors
Thank you for conducting the revisions as per the feedback given.